# Bronchoalveolar Lavage Fluid-Isolated Biomarkers for the Diagnostic and Prognostic Assessment of Lung Cancer

**DOI:** 10.3390/diagnostics12122949

**Published:** 2022-11-25

**Authors:** Alexandros Kalkanis, Dimitrios Papadopoulos, Dries Testelmans, Alexandra Kopitopoulou, Eva Boeykens, Els Wauters

**Affiliations:** 1Department of Respiratory Diseases, University Hospitals Leuven, Campus Gasthuisberg, 3000 Leuven, Belgium; 2Department of Pulmonology, General Hospital Rivierenland, Campus Bornem, 2880 Bornem, Belgium

**Keywords:** biomarkers, bronchoalveolar lavage fluid, lung neoplasm, tumor microenvironment

## Abstract

Lung cancer is considered one of the most fatal malignant neoplasms because of its late detection. Detecting molecular markers in samples from routine bronchoscopy, including many liquid-based cytology procedures, such as bronchoalveolar lavage fluid (BALF), could serve as a favorable technique to enhance the efficiency of a lung cancer diagnosis. BALF analysis is a promising approach to evaluating the tumor progression microenvironment. BALF’s cellular and non-cellular components dictate the inflammatory response in a cancer-proliferating microenvironment. Furthermore, it is an essential material for detecting clinically significant predictive and prognostic biomarkers that may aid in guiding treatment choices and evaluating therapy-induced toxicities in lung cancer. In the present article, we have reviewed recent literature about the utility of BALF analysis for detecting markers in different stages of tumor cell metabolism, employing either specific biomarker assays or broader omics approaches.

## 1. Introduction

Lung cancer is among the most common malignant neoplasms and is associated with a poor prognosis since it is usually diagnosed at an advanced stage [1]. However, recent advances in the field of systemic treatment have changed the landscape of the disease. Much of this progress has been achieved by the discovery and utility of predictive biomarkers, whose detection may guide targeted therapies in an attempt to a more personalized approach to treatment. In cancer research, a biomarker is a biological molecule whose levels can be measured in tissue or fluids and indicate the presence or progress of the disease or its response to specific therapies [2]. Thus, biomarkers may aid in determining the diagnosis, prognosis, and prediction of treatment response in cancer patients. Any component of the cancer cell metabolic pathway, starting from gene alterations and ending with the various metabolites, can be tested as a biomarker according to its biological role. In recent years, the contemporary measurement of multiple substances belonging to the same level in the metabolic cascade has been possible with the use of omics technologies, making feasible the identification of biomarker profiles or signatures [2].

A major problem in employing lung cancer biomarker testing in everyday clinical practice is obtaining adequate tissue samples. Most patients are diagnosed based on small bioptic or even more limited cytological materials obtained during bronchoscopy or transthoracic aspiration. These materials are often depleted during standard histological and histochemical procedures used for diagnosis and tumor subtyping, leaving no space for biomarker testing. Fluids, on the other hand, are promising materials for biomarker evaluation since most substances, from DNA to proteins and metabolites, are being excreted from cancer cells in soluble form. Bronchial or bronchoalveolar lavage fluid (BALF) is obtained by a lung irrigation technique routinely performed during bronchoscopy. It allows the harvesting of cellular and non-cellular contents of the bronchial and alveolar space, serving as an excellent marker of the tumor microenvironment [3]. Its collection requires a minimally invasive procedure that can be repeated with minimum risk, providing a valid and adequate medium for biomarker testing in lung cancer. In addition, due to its proximity to the neoplastic tissue, it could have higher sensitivity for biomarker detection, especially in locally advanced non-metastasized lung cancer, compared to other fluids commonly employed for the same reason, such as plasma or pleural fluid [4].

BALF analysis in lung cancer has multiple uses. Numerous research studies have demonstrated the benefit of employing BALF for the cytological identification of malignant lung neoplasms. Malignant cells may be shed from an adjacent lung tumor into the respective bronchoalveolar space and identified with conventional cytology. Although BALF cytology alone has a modest sensitivity for diagnosing lung cancer (between 29% and 69%), the specificity is exceptionally high (between 90% and 100%) [5]. Furthermore, its combination with other histological techniques seems to increase the diagnostic yield. The diagnostic accuracy of transbronchial lung biopsy in endoscopically non-visible lesions was enhanced when combined with BALF cytology [6]. Secondly, the parenchymal cells of the lungs secrete different types of soluble substances, which can be identified and measured in BALF after removing the cellular component with centrifugation (supernatant). With the advent of novel molecular techniques, it has become possible to amplify nucleic acids that are either extracted from cells or extracellular vesicles (EVs) or detected in cell-free form on BALF and perform a wide array of genomic, epigenomic, and transcriptomic testing. Finally, BALF analysis may aid in identifying various treatment-related adverse events in the lungs, such as pulmonary infections or pneumonitis.

This review aims to perform a brief and comprehensive overview of studies published in the previous decade that focus on the utility of BALF for detecting different forms of diagnostic, predictive, and prognostic biomarkers in lung cancer. Studies were identified by searching the MEDLINE/PubMed electronic database for relative reports dating from 2011 to September 2022. The search strategy included a combination of MeSH terms (biomarkers; bronchoalveolar lavage; lung neoplasm) and keywords (biomarker; [prognostic or predictive] marker; [bronchoalveolar or bronchial or lung] lavage; [bronchial or lung] washing; [lung or pulmonary] [cancer or neoplasm or tumor]). According to our judgement, some older seminal articles are also mentioned based on their influence on the afterward-growing literature on the subject.

## 2. BALF Biomarkers for the Diagnosis and Prognosis of Lung Cancer

### 2.1. Genetic Biomarkers

Recent studies have conducted various assessments on genetic markers at a large, genome-wide scale to identify more specific and sensitive molecules for targeted drug delivery methods to cure the disease. A recent study compared tumor-derived mutations in similar plasma and BALF samples from patients with non-small-cell lung cancer (NSCLC). Researchers discovered that BALF cell-free DNA (cfDNA) testing is more accurate than serum for detecting mutations related to lung cancer [7]. Clinical applications of BALF cfDNA testing include the detection of mutations in affected individuals and, perhaps, the detection of lung adenocarcinoma. It is convenient to identify mutations related to tumors by targeted sequencing of BALF cfDNA, and this method seems more robust compared to plasma screening. Bronchial washings are also a viable and practical choice for genome-wide molecular studies. Actionable mutations in cancer genes were detected with a higher than 90% concordance with respective gene mutations in tumor biopsies [8].

#### 2.1.1. EGFR Mutations

With cfDNA collected through the BALF supernatant, a polymerase chain reaction (PCR) technique may identify functional EGFR mutations. This might be a quick and accurate way to diagnose NSCLC simultaneously using DNA analysis and morphology [9]. The tissue biopsy results are compared to the detectability of activating EGFR mutations in BALF and blood plasma samples in another study [10]. Under this investigation, BALF was found to be significantly more sensitive than liquid biopsy at detecting an EGFR mutation in the same patients (92.3% vs. 38.5%) [10]. A real-time peptide nucleic acid (PNA)-mediated polymerase chain clamping assay was used to detect the T790M EGFR mutation on both BALF and bronchial biopsy specimens of cancer-afflicted persons [11]. In a different, more thorough investigation, 20 patients with lung adenocarcinoma had their EGFR mutations and T790M mutations molecularly tested using cfDNA from BALF. The combination of PNA-mediated clamping PCR and the PANAMutyperTM R EGFR kit with PNA clamping-assisted fluorescence melting curve analysis produced 91.7% concordance with the results from a tumor biopsy [12]. In another study of patients with advanced NSCLC, cytology samples from BALF were used to identify the T790M EGFR mutation. The findings demonstrated that the clinical benefit of osimertinib treatment was predicted by cytology sample EGFR T790M positivity [13]. Isolated EVs from BALF include DNA that can be used for EGFR genotyping by liquid biopsy. Particularly when compared to liquid biopsy cfDNA, BALF EV DNA is tissue specific and incredibly sensitive. Additionally, tissue re-biopsy is less effective than BALF EV DNA at identifying the T790M mutation in individuals who developed resistance to EGFR tyrosine kinase inhibitors (TKIs) [14].

#### 2.1.2. KRAS Mutations

Clinical uses for KRAS mutation analysis in lung disease include its use as a diagnostic indicator for cancer in sputum and BALF samples [15]. In adenocarcinomas of the lung and related sputum and BALF samples, matched KRAS mutations have been found [16]. Furthermore, it has been demonstrated that patients with negative cytological results may benefit from screening for KRAS mutations in BALF cells to help with a lung cancer diagnosis [17]. KRAS and p53 gene mutations in BALF may serve as helpful biomarkers for the diagnosis of peripheral NSCLC, according to the findings of a recent study [18].

#### 2.1.3. ALK Translocations

Reverse transcription-PCR has been used to detect ALK translocations in cytology samples having a 97% concordance with tissue samples [19].

### 2.2. Epigenetic Biomarkers

DNA methylation markers are frequently employed as biomarkers for early diagnosis or recurrence of cancer. DNA methylation occurs when a methyl group is attached to a cytosine base located in a CpG dinucleotide, which may impact gene transcription. Aberrant DNA methylation can reduce the expression of tumor-suppressor genes and allow cancer cells proliferation. Kim et al. discovered that p16, RASSF1A, H-cadherin, and RAR beta gene methylation in tumors may be valuable biomarkers for the early diagnosis of NSCLC in BALF [20]. Moreover, several other studies also reported the potential application value of p16 promoter methylation in sputum for lung cancer diagnosis [21,22]. A large retrospective cohort showed improved diagnostic efficacy of DNA methylation biomarkers (DNA methylation of p16, TERT, WT1, and RASSF1 gene) in cytological bronchial washings evaluation [23]. When lung cancer patients have bronchial aspirates, the Epi pro-Lung BL Reflex Assay is a powerful and practical diagnostic technique for detecting elevated DNA methylation of the SHOX2 gene locus [24]. In comparison to conventional cytology analysis and serum CEA, the methylation analysis of the SHOX2 and RASSF1A panel in BALF using RT-PCR produced better diagnostic sensitivity (81.0%) and specificity (97.4%) [25]. NSCLC could be found by examining the DNA methylation of 7CpGs (TBX15, PHF11, TOX2, PRR15, TFAP2A, HOXA1, and PDGFRA genes) in bronchial washings [26]. A 5-marker (LHX9, GHSR, HOXA11, PTGER4-2, HOXB4-3) methylation model was recently found to have 70% sensitivity and 82% specificity for discriminating malignant from benign pulmonary nodules [27]. According to a meta-analysis, the detection of DNA methylation of the p16^INK4a^ gene in BALF may be a potential biomarker for NSCLC diagnosis [28].

### 2.3. Post-Transcriptional Biomarkers

Transcriptome comparison allows for the discovery of genes that are differently expressed in various cell groups or in response to different treatments. Researchers have showed that clusters of deregulated miRNAs were expressed in the BALF of smoking-related diseases such as lung adenocarcinoma and COPD [29]. Similarly, the expression of two anti-apoptotic genes, survivin and livin, through their mRNA levels was observed in lung cancer patients versus patients with benign lung disease, using bronchial aspirates as a sample source [30]. Other studies have broadened the target transcriptomes to other specific panels of miRNAs that were significantly upregulated in patients with lung cancer [31,32]. Cluster analysis of the expression levels of a miRNA panel in BALF samples from NSCLC patients provided a diagnostic sensitivity of 85.7% and specificity of 100% [33]. Moreover, a prediction model of specific miRNA biomarkers from liquid cytological specimens has shown potential in discriminating squamous cell carcinoma from adenocarcinoma [34]. Transcriptome signatures can also be potentially helpful in predicting survival for patients with lung cancer. Patients with a low expression level of a 3-miRNA panel were associated with better overall survival [35]. Evaluation of EVs from plasma and BALF in patients with NSCLC and benign lung diseases revealed significant differences in exosome amount and miRNA content between fluids and patient groups, revealing their specific role as biomarkers [36]. A significant increase in the expression profile of immunoglobulin genes in BALF was found between lung cancer and healthy controls in a recent transcriptomics study, which generated a 53-gene signature that showed a significant correlation with inhibitory checkpoint PDCD1 [37].

### 2.4. Post-Translational Biomarkers

#### 2.4.1. Proteins

Proteins in BALF can represent the physiological and pathological state of the lung. Proteins that lung cancer cells secrete into BALF may be utilized as biomarkers to detect and evaluate malignancy, aiding lung cancer diagnosis, prognosis, subtyping, and therapy response monitoring. Researchers have since long identified a link between bronchial secretions and serum tumor marker concentrations [38]. Among many proteins, the most specific is napsin A, a renowned immunohistochemical marker for lung cancer diagnosis in affected individuals [39]. In addition, levels of several BALF proteins located in antioxidant, inflammatory, or anti-angiogenic pathways, such as cytokines [40,41,42] and growth [43,44,45], pro-angiogenic [46,47,48], or complement [49,50,51] factors, have been evaluated as diagnostic or prognostic biomarkers employing antibody-specific immunoassays.

Combining proteomics, mass spectrometry (MS), and affinity chemistry-based methods have significantly improved our understanding of the protein oxidative changes that take place in many biological specimens under varied physiological and pathological circumstances in recent years (Figure 1). Glycoproteins are crucial in the standard processing of biological processes such as cell separation, growth, their close contact with their vicinity and invading of tumor cells into surrounding cells. A cohort study used solid-phase N-proteoglycans extraction, iTRAQ labeling, and liquid chromatography-tandem MS to analyze N-proteoglycans levels in BALF from patients with lung cancer and benign lung diseases. Levels of 8 glycoproteins (neutrophil elastase, integrin alpha-M, cullin-4B, napsin A, lysosome-associated membrane protein 2, cathepsin D, BPI fold-containing family B member 2, and neutrophil gelatinase-associated lipocalin) displayed more than two times rise in cancer BALF compared to benign BALF [52]. According to research by Uribarri et al. the expression levels of APOA1, CO4A, CRP, GSTP1, and SAMP led to a diagnostic panel for lung cancer that was 95% sensitive and 81% specific. The measurement of STMN1 and GSTP1 proteins enabled the two main subtypes of lung cancer (non-small-cell and small-cell) to be distinguished with 57% specificity and 90% sensitivity [53]. Ortea et al. identified different proteins in the BALF of patients with lung cancer, including glutathione S-transferase pi, haptoglobin, and complement C4-A. These different types of proteins could be proved as prominent biomarkers for disease diagnosis [54]. Almatroodi et al. performed a quantitative proteomic evaluative study on BALF of patients with lung adenocarcinoma, and their results indicated the occurrence of 1100 proteins in BALF. Among them, the ratio of over-expressed proteins in lung adenocarcinoma individuals were 33 compared to healthy individuals. S100-A8, thymidine phosphorylase, annexin A2, transglutaminase 2, and annexin A1 were lung cancer individuals’ most renowned over-expressed proteins [55]. In another prospective cohort of individuals with suspected lung cancer, proteomic analysis of BALF samples identified 133 proteins that could differentiate cancer and non-cancer patients [56]. Using label-free MS analysis of BALF, Hmmier et al. discovered that 4 proteins (cystatin-C, TIMP1, lipocalin 2, and HSP70/HSPA1A) were significantly overexpressed in lung cancer patients compared to healthy controls [57]. A recent investigation evaluated aberrant protein glycosylation using lectin microarrays in BALF. It revealed 15 lectins that could distinguish between the different lung cancer types and 14 lectins whose levels differed between early and advanced disease stages [58].

#### 2.4.2. Cell Epitopes

The combination of immunoassays and flow cytometry in BALF supernatant enables the study of cell epitopes, mainly used in identifying immune cell subpopulations (Figure 2). T cells are essential for antitumor defense, but as cancer progresses, their population changes. Numerous suppressory and regulatory systems can prevent lymphocyte activation and prevent the identification of lung cancer antigens. Various research studies indicated that now lung cancer immunotherapeutic therapy aims to enhance the cytotoxological action of lymphocytes by restricting the activities of suppressor molecules such as cytotoxic T cell antigen 4 (CTLA-4) and programmed death-1 (PD-1) [59]. The examination of BALF revealed that one could easily analyze the different types of immune responses in the tumor microenvironment as well as an alteration in the status of various cells. A higher ratio of cytotoxic CD8+ lymphocytes, neutrophils, T cells, and CTLA-4+ T regulatory cells was found in BALF of the cancer-affected lung compared to the healthy contralateral lung [60]. Moreover, tissue samples of lung adenocarcinoma patients also indicated the presence of M2 polarization of macrophages, which are further associated with the onset of lung cancer [61]. Osińska et al. examined the BALF of 35 patients with lung cancer to figure out the existence of regulatory T cells (Tregs) and their specific role in boosting immunity. Flow cytometry analysis revealed that Treg cells were present in significant proportion in the local environment of lung cancer cells as compared to normal cells. These Tregs are renowned as anticancer defense cells because they have the potential to restrict the normal functioning of NK cells, T-lymphocytes, and dendritic cells, which ultimately enhances the immune tolerance level [62]. In 2018, Hu et al. examined the significant ratio of PD-1+ cells in BALF and peripheral blood in small-cell lung cancer (SCLC) individuals, and their levels dropped after chemotherapy. Therefore, these cells would be potential biomarkers for the diagnosis of diseases as these programmed death cells are recognized as checkpoints in the immune system, and their inhibition could potentially mediate the activation of T cells, ultimately exhibiting antitumor activity [63]. Moreover, another rational study was carried out on BALF as a prognostic agent for patients of lung cancer, and through a cytometric bead array, Hu et al. identified a higher level of interleukin (IL)-10 and IL-10+CD206+CD14+M2-like macrophages in the SCLC affected individuals as compared to NSCLC individuals. These markers were also correlated with advanced stage and reduced survival of SCLC patients [64]. Recently, a cohort study was conducted by Masuhiro et al. to examine the immune profile of tumor microenvironment through BALF of NSCLC individuals according to their response to immunotherapy. They observed a higher level of CXCL9 and a higher ratio of CD56+ cytotoxic T cells in BALF of patients responding to nivolumab compared to those not responding [65].

Some cells disassociate themselves from the primary tumor mass and enter the circulatory system. These cells are known as circulating tumor cells (CTCs) and emerge as a new target that offers clinical insights for the prediction and diagnosis of different types of cancers [66]. Recent studies have demonstrated that the precision and sensitivity of CTCs detected by the chromosomal enumeration probe 8 (CEP8) in lung cancer patients was, correspondingly, 83.3% and 98.6% [67]. Several studies have investigated these distinct CEP8+ CTCs, and the results have demonstrated a substantial correlation between these cells with the detection of lung cancer and also its prognosis with excellent precision and sensitivity [68,69]. Detection of CTCs in serum and BALF showed that CEP8+ CTCs could be used as a supplementary approach for individuals with solitary pulmonary nodules to diagnose lung cancer at an early stage [70].

### 2.5. Metabolite Biomarkers

Metabolites of tumor cells, such as amino acids, nucleic acids, lipids, and sugars, may reflect cellular metabolic processes that drive tumor formation and progression. Either by using metabolite-specific immunoassays or MS-based metabolomic approaches (Figure 3), recent studies have compared metabolites between lung cancer patients and controls. In an early study, levels of leukotriene B4 and cysteinyl leukotrienes, which are end products of arachidonic acid metabolism, were significantly elevated in BALF of NSCLC patients [71]. In another report, patients with NSCLC had significantly lower ATP and ADP concentrations in BALF than patients with COPD, an effect that was exerted by upregulated expression of P2Y1, P2X4, and P2X7 purinergic receptors [72]. Callejón-Leblic et al. identified 42 altered metabolites in BALF of lung cancer patients compared to non-cancer patients. Based on ROC curve analysis, levels of carnitine, adenine, choline, glycerol, and phosphoric acid show high sensitivity and specificity to distinguish between lung cancer and controls and should be considered as potential biomarkers [73].

Certain elements such as essential and non-essential metals (Fe, Cd, Zn, Co, Cu, Pb, V, Cr, etc.) are important for maintaining homeostasis and are involved in several cellular pathways. Excess or deficiency of such elements causes dyshomeostasis and disturbs cellular pathways, ultimately leading to the progression of serious illness [74]. So, identification of levels of such elements in body fluids such as serum and BALF can help to understand the mechanisms involved in the progression of serious illnesses such as lung cancer. As a result, variations in the particular elements’ concentrations and the profile of their chemical components can reveal the individual’s metabolic and nutritional status and therefore offers a prospective early prognosis of cancer progression. Belén et al. used triple quadrupole inductively coupled plasma MS analysis and identified the presence of metals in body fluids, including BALF, in lung cancer patients and concluded that such metals could be employed as a potent biological marker for more accurate and expeditious identification of lung adenocarcinoma in affected individuals [75].

### 2.6. Metagenomic Biomarkers

Microbiome generally contributes towards barrier formation, external communication, and immune homeostasis in healthy individuals. The microbiome can also trigger host immune responses against cancer cells. Several studies have shown the interconnectedness between lung cancer and the microbiome. For example, Wang et al. conducted a preliminary analysis of microbiome diversity in saliva and BALF of lung cancer patients [76]. They observed that the microbiome diversity is significantly less than in the healthy samples, indicating a link between cancer and microbiome. It can also be said that certain microbiome genera could act as potential lung cancer biomarkers. In this regard, Cheng et al. characterized microbiomes in BALF for potential associations with lung cancer [77]. The microbiome diversity was compared between diseased and healthy samples using principal coordinate analysis. They observed six different genera that were significantly enriched in BALF of cancer patients: Blautia, Capnocytophaga, Gemmiger, Oscillospira, Sediminibacterium, and TM7-3. Similarly, Lee et al. investigated the microbiome in BALF of the affected group and compared them with benign lung mass [78]. They reported that members of the genera Megasphaera and Veillonella are potential cancer markers as their concentrations are relatively higher in cancer patients than in the control group. In addition, Patnaik et al. observed that the lower airway microbiome could contribute to the recurrence of early-stage NSCLC [79]. They observed that the microbiome diversity differed significantly between patients with recurrence and non-recurrence after surgery. They also observed that the diversity among 16 out of 18 cancer recurrence patients was similar. Zheng et al. performed metagenomic sequencing on BALF samples from cancer patients and healthy controls [80]. Even though the diversity between both samples was comparable, some rare microbiota stood out in lung cancer samples. For example, species such as Bacteroides pyogenes, Chaetomium globosum, Lactobacillus rossiae, Magnetospirillum gryphiswaldense, Paenibacillus odorifer, and Pseudomonas entomophila were common in cancer samples. The results indicate that change in the microbiome’s diversity is directly associated with cancer. Moreover, Veillonella was found to increase tumor burden and decrease survival. Furthermore, the microbiome plays a crucial role in cancer cells’ response to immunotherapy. For example, Jang et al. analyzed the microbiome’s relationship with programmed death-ligand-1 (PD-L1) expression [81]. PD-L1 protein is typically involved in protecting non-cancerous/non-harmful cells from immune cells. The authors also found that the cells with higher PD-L1 expression levels respond better to immunotherapy than the cells with lower PD-L1 expression levels. The population of Veillonella was significantly higher in the group with the higher expression level of PD-L1, while Neisseria was abundant in the group with the lower expression of PD-L1. Moreover, the exact reason for a higher amount of Veillonella in cancer patients is still unknown, but several studies have developed links between different lung-related diseases and the microbiome genera under debate. For example, Veillonella has been found to be involved in pulmonary fibrosis and asthma development [82,83]. As both pulmonary disorders are related to the immune system and lungs, it is also expected that Veillonella will be higher in cancer patients as well.

## 3. BALF Biomarkers for the Identification of Adverse Events of Lung Cancer Treatment

### 3.1. Pulmonary Infections

It has been investigated that the risk of developing neutropenia is usually low in the case of lung adenocarcinoma immunotherapeutic procedures compared to chemotherapy [84]. Moreover, long-term immunosuppression therapies with high drug dosage (steroids, TNF-α) are required for immune-related adverse events (irAEs), e.g., colitis and pneumonitis, that ultimately enhance the risk of other infections [85]. Aspergillus fumigatus pneumonia diagnosed after BALF examination has been described in a metastatic melanoma patient treated with systemic corticosteroids and infliximab for an irAE due to ipilimumab [86]. Another report described the detection of Mycobacterium tuberculosis in BALF in a NSCLC patient during third-line nivolumab therapy [87]. Thus, the performance of bronchoscopy and subsequent serological and molecular testing of BALF is an established method for the diagnosis of such infectious complications in this unique patient population [88].

### 3.2. Therapy-Induced Pulmonary Toxicity

It has been observed that excess application of immunotherapeutic molecules has important side effects; among them, checkpoint inhibitor pneumonitis (CIP) is the most famous. Common clinical presentations of CIP are the onset of dyspnea, hypoxemia, and pulmonary infiltration [89,90]. For advanced NSCLC, the phase I trial of pembrolizumab and nivolumab reported 1 and 3 pneumonitis-related deaths, respectively [91,92]. The extensive examination of BALF showed infiltration and inflammation of lymphocytes in patients with CIP. In 2017, another case study reported T-lymphocytic alveolitis as extrinsic allergic alveolitis in 80% of affected individuals with CD8+ cell predominance [93]. Identifying the dysregulation of immune cell subpopulations in BALF could serve as a promising target for developing therapeutics to minimize adverse effects [94].

Drug-induced sarcoidosis reaction (DISR) can be another after-effect of cancer-related therapies. DISR is a systematic granulomatous reaction that arises after using chemotherapeutic agents and is indistinguishable from sarcoidosis [95]. DISR has been described after treatment with immune checkpoint inhibitors, including PD-L1 inhibitors, PD-1 inhibitors, and anti-CTLA-4 antibodies [96]. Thus, in the absence of biopsy specimens, the surrogate method for the diagnosis of DISR in affected patients is BALF sampling [97]. Montaudie et al. conducted research work, and they presented a DISR-based study that showed 32% lymphocytes along with the enhanced frequency of CD4/CD8 cells [98].

Radiation-induced pneumonitis is an inflammatory process occurring in reaction to chest radiotherapy used to treat thoracic malignancies. Early identification and treatment are crucial, considering the fact that its chronic phase constitutes lung fibrosis. Previous research has detected different types of serum biological markers, including TGF-β, IL-6, TNF-α, and IL-1, that predicted tissue injuries after radiotherapy [99,100]. Later, a series of research studies were performed by Kwang-Joo Park et al. on animal models for the evaluation of radiotherapy-associated injuries of lung tissues, in which the focus was on BALF. Their results revealed that the level of IL-1, NO, TGF-β, and neutrophils were high in BALF [101,102]. Additionally, lymphocytic alveolitis was also found in BALF of affected individuals who took radiotherapy for breast cancer. In most symptomatic cases, a significant rise in CD4+ T lymphocytes was observed, but no increase in CD4/CD8 ratio was found [103]. Crohns et al. reported that radiation therapy resulted in a significant increase in IL-6 in BALF, while elevated IL-8 levels in BALF are associated with worse survival in lung cancer patients receiving radiotherapy [104]. BALF was found to be more effective as compared to serum for the detection of surfactant protein D, which is an important biomarker for radiation-induced pneumonitis in tumor tissues of affected persons [105]. In a cohort study, the profile of cytokines or chemokines was analyzed in BALF samples from cancer-afflicted individuals before and after radiotherapy. This study showed the overexpression of cytokines, including PAI-1, CD154, IL-1ra, CXCL-1, IL-23, MIF, and IFN-γ, in the lungs of patients with high-grade radiation-induced pneumonitis even before radiotherapy [106].

## 4. The Value of Repeated BALF Examination in the Course of Lung Cancer

With the advent of modern molecular technologies and extensive research work in biological sciences, highly curable therapies for patients with lung cancer have been recently introduced. These treatment methods wrapped new therapeutic molecules as well as more specific biomarkers for the diagnosis of cancer tissues found in different sections of the lungs more accurately. Apart from their highly beneficial results, the continuous re-evaluation of cancer patients is necessary to gauge the tumor status before and after the application of each line of therapy. It has been proposed that the follow-up should not only be restricted to imaging but repeated histological assessment may be needed when relapses occur after therapeutic interventions. Zarogoulidis et al. performed a case study, and their results suggested that re-biopsy of different lung cancer tissues after the administration of TKIs would detect T790M mutations as well as an altered type of lung cancer [107]. Another cohort study reported that 63% of NSCLC patients showed onset of EGFR mutations and proliferation of cancer after treatment with EGFR-TKIs during re-biopsy, while a 33% frequency of T790M mutation was found [108]. Taking into account the low invasiveness and high sensitivity of BALF testing, a surrogate method of the classic re-biopsy technique could be repeated BALF assessment, not only for the detection of T790M mutations but also for a re-evaluation of biomarkers that may predict cancer recurrence or prognosis and guide further treatment choices.

## 5. Advantages and Disadvantages of BALF Biomarker Testing in Lung Cancer

Biomarker testing is a central component in nowadays personalized management of lung cancer patients. Many approved drugs for the treatment of stage IV NSCLC are being prescribed according to the detection and quantification of specific molecular markers, such as EGFR, ALK, ROS1, and BRAF mutations and PD-L1 expression. In the case of minimal or no tumor tissue material available for biomarker testing, BALF has the advantage of containing a wide variety of molecules, either cell derived or in soluble form, that resemble the tumor molecular profile. Biomarkers with the highest diagnostic, predictive, and prognostic accuracy from the studies reported above are presented in Table 1. High concordance to tissue markers and better accuracy than plasma in oncogene driver detection makes BALF the best alternative for predictive assessment in lung neoplasms. Panels of BALF-isolated biomarkers of pre- and post-transcriptional gene expression regulation, such as DNA methylation and miRNA expression, have higher diagnostic accuracy than BALF cytology for the detection of lung cancer. Moreover, prediction models derived from proteomic, metabolomic, and microbiomic studies have identified novel markers for the discrimination between cancer types and stages and the prediction of immunotherapy response or tumor progression/recurrence.

Main disadvantages of BALF biomarker testing are the lower yield of tumor cells or tumor nucleic acids compared to tissue samples, which could impede biomarker test performance, and the higher invasiveness compared to plasma sampling. Furthermore, continuous validation of the methylomic, transcriptomic, and proteomic signatures in BALF is needed so that they may assist in the diagnosis of small pulmonary nodules and guide treatment approaches in clinical practice.

## 6. Conclusions

It can be concluded that BALF is an appropriate medium that may aid in the diagnosis of lung cancer, in assessing prognosis and response to therapy, but also in the early identification of treatment-related adverse events. Furthermore, continuous re-evaluation of the genome and the protein and immune status of lung cancer cells is essential in the course of the disease. Repeated BALF examination is a cost-effective and easily accessible method for evaluating the microenvironment and immune status of cancer-afflicted lung tissues and provides a valid comparison of the disease status before and after treatment.

## Figures and Tables

**Figure 1 diagnostics-12-02949-f001:**
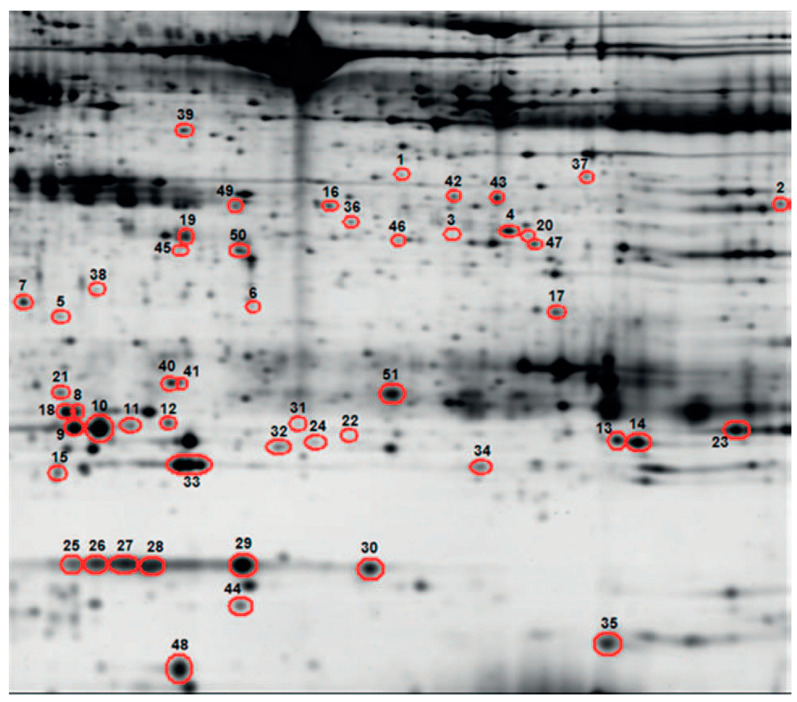
Example of a proteome from a bronchoalveolar lavage fluid sample in a lung cancer patient performed by two-dimensional gel electrophoresis and silver staining. Circled spots represent differentially expressed proteins between lung cancer and controls. Reprinted from [53] with permission from Elsevier.

**Figure 2 diagnostics-12-02949-f002:**
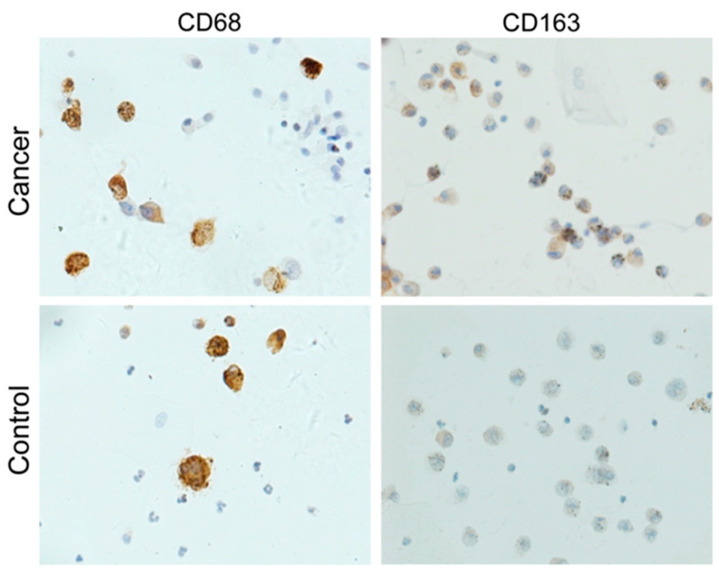
Immunostaining for cell epitopes in macrophages from bronchoalveolar lavage fluid samples in a lung cancer patient and healthy control. Reprinted from [48] with permission from Elsevier.

**Figure 3 diagnostics-12-02949-f003:**
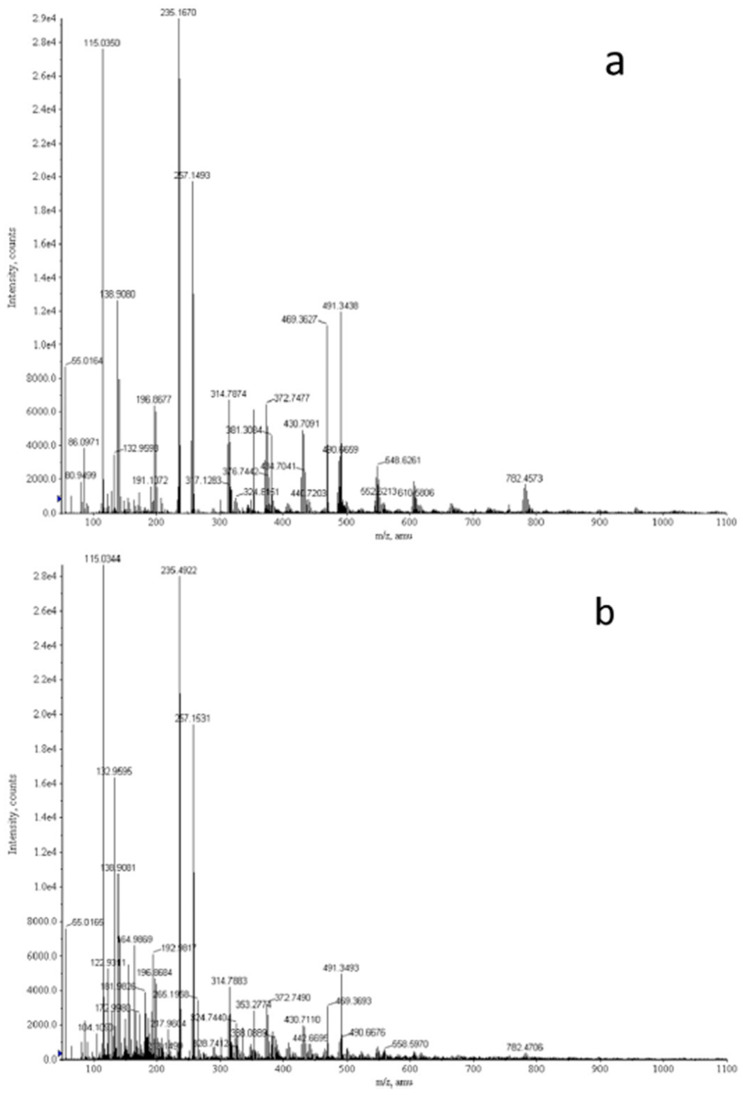
Typical mass spectra from lung cancer patients (**a**) and controls (**b**) by direct infusion triple quadrupole time-of-flight mass spectrometry in bronchoalveolar lavage fluid for the detection of metabolite expressions. Reprinted from [73] with permission from Elsevier.

**Table 1 diagnostics-12-02949-t001:** Overview of lung cancer biomarkers, their measurement methods in bronchoalveolar lavage fluid, their respective diagnostic/predictive roles, and accuracy results from studies included in the review.

Study	Patients	Controls	Processing	Detection	Biomarker	Role	Accuracy
Genetic/epigenetic biomarkers							
Lee 2020 [4]	*n* = 73 (M 38) Mean age: 65.3 ± 9.8 y Type: AC 65, SQCC 7, other 1 Stage: I 20, II 10, III 10, IV 27	-	Cell-free DNA from BALF supernatant	Droplet digital allele-specific PCR	EGFR L858R mutation	Prediction of tumor molecular profile	AUC 0.96 Acc 95%
					EGFR E19del mutation		AUC 0.86 Acc 85%
Nair 2022 [7]	*n* = 35 (M 18) Age: 40–83 y Type: AC 30, SQCC 2, NSCLC NOS 1, SCC 1, other 1 Stage: I 18, II 6, III 6, IV 5	*n* = 21 (M 12) Age: 46–76 y	Cell-free DNA from BALF supernatant	Deep sequencing	11 gene feature classifier	LC diagnosis	AUC 0.84 Sens 69% Spec 100%
	*n* = 31	-			Any tumor variant detected	Prediction of tumor molecular profile	Acc 81%
Roncarati 2020 [8]	*n* = 91 (M 60) Age: 47–85 y Type: AC 41, SQCC 31, SCC 11, undefined 7 Stage: I 13, II 7, III 25, IV 43	*n* = 31 (M 21) Age: 42–86 y	Cellular DNA/RNA from BALF cell pellet	Droplet digital methylation-specific PCR	CDH1 methylation	LC diagnosis	Sens 64% Spec 74%
					DLC1 methylation		Sens 37% Spec 94%
					PRPH methylation		Sens 40% Spec 100%
					RASSF1A methylation		Sens 46% Spec 100%
					4-gene methylation panel		AUC 0.93 Sens 97% Spec 74%
		-		Next-generation sequencing	ALK fusions	Prediction of tumor molecular profile	Acc 96%
					BRAF V600E mutation		Acc 100%
					EGFR mutations		Acc 97%
					ERBB2/HER2 mutations		Acc 100%
					KRAS mutations		Acc 90%
					MET mutations		Acc 100%
					ROS1 fusions		Acc 100%
Kawahara 2015 [9]	*n* = 42 (M 23) Age: 42–84 y Type: AC	-	Cell-free DNA from BALF supernatant	Allele-specific PCR/ FRET-PHFA	EGFR mutations	Prediction of tumor molecular profile	Acc 47%
Yanev 2021 [10]	*n* = 26 (M 13) Mean age: 63.3 y Type: AC	-	Cell-free DNA from BALF supernatant	Allele-specific PCR	EGFR mutations	Prediction of tumor molecular profile	Acc 92%
Park 2017 [12]	*n* = 20 (M 5) Age: 43–77 y Type: AC Stage: I 1, II 1, III 1, IV 17	-	Cell-free DNA from BALF supernatant	PNA-mediated clamping PCR/ PNA clamping-assisted fluorescence melting curve analysis	EGFR mutations	Prediction of tumor molecular profile	Acc 92%
						Prediction of treatment response (EGFR TKI)	-
Hur 2018 [14]	*n* = 23	-	EV DNA from BALF EV pellet	PNA-mediated clamping PCR	EGFR mutations	Prediction of tumor molecular profile	Acc 100%
			Cell-free DNA from BALF supernatant				Acc 71%
Ahrendt 1999 [16]	*n* = 50 Type: AC 25, SQCC 23, other 2 Stage: I 28, II 15, III 7	-	Cellular DNA from BALF cell pellet	Oligonucleotide plaque hybridization	p53 mutations	Prediction of tumor molecular profile	Acc 39%
				Allele-specific PCR	KRAS mutations		Acc 50%
				Methylation-specific PCR	p16 methylation		Acc 63%
				Microsatellite fragment analysis	15 microsatellite markers		Acc 14%
					Combined panel		Acc 53%
Oshita 1999 [17]	*n* = 20 Type: AC	*n* = 13	Cellular DNA from BALF cell pellet	Allele-specific PCR	KRAS codon 12 mutation	LC diagnosis	Sens 79% Spec 64%
Li 2014 [18]	*n* = 48 (M 27) Age: 42–75 y Type: AC 32, SQCC 14, LCC 2 Stage: I 19, II 25, III 4	*n* = 26 (M 17) Age: 28–70 y	Cellular DNA from BALF cell pellet	PCR single-strand conformation polymorphism	KRAS mutations	LC diagnosis	Sens 38% Spec 92%
						Prediction of tumor molecular profile	Acc 72%
					p53 mutations	LC diagnosis	Sens 44% Spec 96%
						Prediction of tumor molecular profile	Acc 75%
					Combined panel	LC diagnosis	Sens 67% Spec 89%
						Prediction of tumor molecular profile	Acc 70%
Nakamichi 2017 [19]	*n* = 36	-	Cellular RNA from BALF cell pellet	mRNA-specific reverse transcription-PCR	ALK translocations	Prediction of tumor molecular profile	Acc 97%
Kim 2004 [20]	*n* = 85 (M 57) Mean age: 65 ± 17 y Type: AC 31, SQCC 43, other 11 Stage: I 52, II 33	*n* = 127 (M 84) Mean age: 62 ± 14 y	Cellular DNA from BALF cell pellet	Methylation-specific PCR	p16 methylation	LC diagnosis	Sens 16% Spec 94%
					RARβ methylation		Sens 15% Spec 87%
					H-cadherin methylation		Sens 13% Spec 97%
					RASSF1A methylation		Sens 18% Spec 96%
Nikolaidis 2012 [23]	*n* = 139 (M 80) Mean age: 68.4 ± 8.1 y Type: AC 22, SQCC 31, SCC 39, LCC 16, other 20, unknown 11	*n* = 109 (M 63) Mean age: 67.6 ± 8.8 y	Cellular DNA from BALF cell pellet	Methylation-specific PCR	Methylation panel (p16, RASSF1, WT1, TERT)	LC diagnosis	Sens 82% Spec 91%
Dietrich 2012 [24]	*n* = 125 (M 72) Age: 46–85 y Type: AC 26, SQCC 28, NSCLC NOS 9, SCC 40, other 22	*n* = 125 (M 61) Age: 45–86 y	Cellular DNA from BALF cell pellet	Methylation-specific PCR	SHOX2 methylation	LC diagnosis	AUC 0.94 Sens 78% Spec 96%
Zhang 2017 [25]	*n* = 284 (M 212) Age: 31–85 Type: AC 92, SQCC 107, SCC 42, LCC 5, unknown 38 Stage: I 28, II 30, III 133, IV 93	*n* = 38 (M 28) Age: 29–75 y	Cellular DNA from BALF cell pellet	Methylation-specific PCR	Methylation panel (SHOX2, RASSF1A)	LC diagnosis	AUC 0.89 Sens 81% Spec 97%
Um 2018 [26]	*n* = 31	*n* = 10	Cellular DNA from BALF cell pellet	DNA methylation microarray	Methylation panel (TFAP2A, TBX15, PHF11, TOX2, PRR15, PDGFRA, HOXA11)	LC diagnosis	AUC 0.87 Sens 87% Spec 83%
Li 2021 [27]	*n* = 52 (M 33) Type: AC 37, SQCC 10, other 5 Stage: I 34, II 1, III 1, IV 4, unknown 12	*n* = 59 (M 33)	Cellular DNA from BALF cell pellet	Methylation-specific PCR	Methylation panel (LHX9, GHSR, HOXA11, PTGER4-2, HOXB4-3)	LC diagnosis	AUC 0.82 Sens 70% Spec 82%
Post-transcriptional biomarkers							
Molina-Pinelo 2014 [29]	*n* = 48 (M 40) Type: AC Stage: I-II 2, III 15, IV 31	*n* = 16 (M 15)	Cellular RNA from BALF cell pellet	MicroRNA microfluidic card array	Four upregulated miRNA clusters (chromosome loci 13q31.3, 7q22.1, Xq26.2, 11q13.1)	LC diagnosis	-
Li 2013 [30]	*n* = 70 (M 52) Mean age: 64 ± 24 y Type: AC 37, SQCC 25, LCC 4, SCC 4 Stage: I 10, II 24, III 25, IV 11	*n* = 26 (M 19) Mean age: 55 ± 19 y	Cellular RNA from BALF cell pellet	mRNA-specific reverse transcription-PCR	Survivin expression ratio > 0.35	LC diagnosis	AUC 0.83 Sens 83% Spec 96%
					Livin expression ratio > 0.3		AUC 0.68 Sens 63% Spec 92%
Rehbein 2015 [31]	*n* = 30 (M 21) Median age 64.5 y	*n* = 30 (M 17) Median age 63.5 y	Cell-free RNA from BALF supernatant	MicroRNA microfluidic card array	Five upregulated miRNAs (U6 snRNA, hsa-miR 1285, hsa-miR 1303, hsa-miR 29a-5p, hsa-miR 650)	LC diagnosis	-
Kim 2018 [32]	*n* = 13 (M 7) Age: 47–72 y Type: AC Stage: I 10, II 3	*n* = 15	EV RNA from BALF EV pellet	miRNA-specific reverse transcription-PCR	miR-126 and Let-7a were significantly upregulated	LC diagnosis	-
Kim 2015 [33]	*n* = 21 (M 17) Age: 46–84 y Type: AC 13, SQCC 5, LCC 3 Stage: I 12, II 9	*n* = 10 (M 8) Age: 30–77 y	Cellular RNA from BALF cell pellet	miRNA-specific reverse transcription-PCR	High expression cluster of a 5-miRNA panel (miR-21, miR-143, miR-155, miR-210, miR-372)	LC diagnosis	Sens 86% Spec 100%
Li 2017 [34]	*n* = 127 (M 82) Age: 66 ± 8 y Type: AC 45, SQCC 82 Stage: I 52, II 38, III-IV 37	-	Cellular RNA from BALF cell pellet	Droplet digital miRNA-specific PCR	2-miRNA (miR-205-5p, miR-944) prediction model	Discrimination of SQCC from AC	AUC 0.997 Sens 95% Spec 97%
Mancuso 2016 [35]	*n* = 50 (M 32) Age: 34–82 y Type: SCC Stage: III 18, IV 32	-	Cellular RNA from BALF cell pellet	miRNA-specific reverse transcription-PCR	Above median expression levels of a 3-miRNA panel (miR-192, miR-200c, miR-205)	Overall survival (worse)	-
Rodríguez 2014 [36]	*n* = 30 (M 23) Age: 45–83 y Type: AC 14, SQCC 16	*n* = 75 (M 46) Age: 18–87 y	EV RNA from BALF EV pellet	MicroRNA real-time PCR array	10 miRNAs were upregulated and 10 downregulated	LC diagnosis	-
Kuo 2018 [37]	*n* = 34 (M 19) Mean age: 58.5 ± 12.8 y Type: AC 26, SQCC 8 Stage: III 11, IV 23	*n* = 14 (M 7) Mean age: 53.3 ± 11.4 y	Cellular RNA from BALF cell pellet	mRNA-specific reverse transcription-PCR	9-gene (SPP1, CEACAM6, MMP7, SLC40A1, IGJ, IGKC, CPA3, YES1, CXCL13) prediction model	LC diagnosis	AUC 0.92
Post-translational biomarkers (proteins, cell epitopes, metabolites)							
Macchia 1987 [38]	*n* = 37 Type: AC 4, SQCC 23, LCC 3, SCC 7	*n* = 20	Cell-free BALF supernatant	RIA	CEA	LC diagnosis	Sens 57% Spec 65%
					TPA		Sens 65% Spec 20%
					NSE	SCC diagnosis	Sens 71% Spec 90%
					Ferritin		Sens 71% Spec 100%
					CanAg CA-50		Sens 100% Spec 55%
Naumnik 2012 [40]	*n* = 45 (M 38) Mean age: 61.9 ± 4 y Type: AC 9, SQCC 22, NSCLC NOS 14 Stage: III 18, IV 27	*n* = 15 (M 13) Mean age: 60.1 ± 5 y	Cell-free BALF supernatant	ELISA	IL-27 (↑)	LC diagnosis	-
					IL-27, IL-29 (↓)	Discrimination of advanced stage	-
Naumnik 2016 [41]	*n* = 46 (M 46) Mean age: 63 ± 3 y Type: AC 10, SQCC 25, LCC 11 Stage: III 20, IV 26	*n* = 15 (M 12) Mean age 60 ± 4 y	Cell-free BALF supernatant	ELISA	HGF, IL-22 (↓)	LC diagnosis	-
					IL-22 (↑)	Overall survival (worse)	-
Kontakiotis 2011 [42]	*n* = 42 (M 42) Age: 43–80 y Type: AC 7, SQCC 22, SCC 10, other 3	*n* = 16 (M 16) Age: 45–77 y	Cell-free BALF supernatant	ELISA	TNF-α (↑)	LC diagnosis	-
				Colorimetric assay	Total antioxidants, glutathione (↑)		-
Jakubowska 2015 [43]	*n* = 45 (M 38) Mean age: 61.7 ± 8.3 y Type: AC 20, SQCC 22, LCC 3 Stage: III 18, IV 27	*n* = 15 (M 13) Mean age: 60.1 ± 5.0 y	Cell-free BALF supernatant	ELISA	TGF-β (↑)	LC diagnosis	-
Chen 2014 [44]	*n* = 45 (M 28) Mean age: 60.8 ± 1.2 y Type: AC 11, SQCC 18, SCC 10, other 6	*n* = 33 (M 19) Mean age: 58.2 ± 1.7 y	Cell-free BALF supernatant	ELISA	TGF-β1 >10.85 pg/ml	LC diagnosis	AUC 0.7 Sens 62% Spec 61%
Xiong 2020 [45]	*n* = 219 (M 150) Mean age: 68.4 ± 18.8 y Type: AC 136, SQCC 43, SCC 35, other 5 Stage: 0 38, I 93, II 50, III 28, IV 10	*n* = 186 (M 125) Mean age: 40.6 ± 15.5 y	Cell-free BALF supernatant	ELISA	VEGF >234.1 pg/mL, TGF-β >81.8 pg/mL, HGF 44.6 pg/mL (at least 2 positive)	LC diagnosis	AUC 0.81 Sens 82% Spec 61%
Charpidou 2011 [46]	*n* = 40 (M 37) Age: 45–82 y Type: AC 12, SQCC 19, other 9 Stage: I 3, III 14, IV 23	-	Cell-free BALF supernatant	ELISA	VEGF (↓)	Prediction of treatment response (chemotherapy)	-
					VEGFR1 >53.2 pg/ml	Progression free survival (worse)	-
					VEGFR2 >705.3 pg/ml	Overall survival (worse)	-
Cao 2013 [47]	*n* = 37 (M 28) Mean age: 55.4 ± 8.4 y Type: AC 15, SQCC 19, SCC 3 Stage: I 23, II 9, III 5	*n* = 19 (M 12) Mean age: 48.1 ± 9.2 y	Cell-free BALF supernatant	ELISA	VEGF >214 pg/ml	LC diagnosis	AUC 0.86 Sens 82% Spec 84%
Chen 2014 [48]	*n* = 54 (M 60) Median age: 60 y Type: AC 9, SQCC 36, SCC 9	*n* = 12 (M 6) Median age: 37 y	Cell-free BALF supernatant	ELISA	IL-8, VEGF (↑)	LC diagnosis	-
Pio 2010 [49]	*n* = 56 (M 45) Age: 38–83 y Type: AC 12, SQCC 24, LCC 4, SCC 9, other 7	*n* = 22 (M 14) Age: 30–82 y	Cell-free BALF supernatant	ELISA	Complement factor H >1 μg/mL	LC diagnosis	Sens 62% Spec 77%
					Albumin >17 μg/mL		Sens 68% Spec 71%
Ajona 2013 [50]	*n* = 50 (M 41) Type: AC 12, SQCC 22, SCC 9, other 7	*n* = 22 (M 14)	Cell-free BALF supernatant	ELISA	Complement C4-derived fragments	LC diagnosis	AUC 0.73
Ajona 2018 [51]	*n* = 49 (M 40) Type: AC 12, SQCC 22, SCC 8, other 7	*n* = 22 (M 14)	Cell-free BALF supernatant	ELISA	Complement C4d	LC diagnosis	AUC 0.80
Li 2013 [52]	*n* = 18 (M 7) Age: 51–83 y Type: AC	*n* = 6 (M 3) Age: 18–85 y	Cell-free proteins from BALF supernatant	ELISA	Napsin A >55 ng/mg total protein	LC diagnosis	AUC 0.85 Sens 84% Spec 67%
Uribarri 2014 [53]	*n* = 204 (M 177) Mean age: 63.0 ± 10.7 y Type: AC 59, SQCC 80, SCC 63, other 2 Stage: I 14, II 4, III 60, IV 109, undefined 17	*n* = 48 (M 38) Mean age: 54.9 ± 14.0 y	Cell-free proteins from BALF supernatant	Fluorescent bead-based immunoassay	5-protein (APOA1, CO4A, CRP, GSTP1, SAMP) prediction model	LC diagnosis	AUC 0.94 Sens 95% Spec 81%
					2-protein (STMN1, GSTP1) prediction model	Discrimination of SCC from NSCLC	AUC 0.80 Sens 90% Spec 57%
Ortea 2016 [54]	*n* = 12 (M 8) Median age: 64 y Type: AC Stage: I-II 2, III-IV 10	*n* = 10 (M 10) Median age: 61	Cell-free proteins from BALF supernatant	Liquid chromatography–mass spectrometry	Discriminant analysis of a 44-protein panel	LC diagnosis	Sens 92% Spec 70%
Almatroodi 2015 [55]	*n* = 8 (M 5) Mean age: 68.1 ± 7.6 y Type: AC Stage: I 2, II 2, III 1, IV 3	*n* = 8 (M 3) Mean age: 60 ± 8.7 y	Cellular proteins from BALF cell pellets	Liquid chromatography–mass spectrometry	33 upregulated proteins	LC diagnosis	-
Carvalho 2017 [56]	*n* = 49 Type: AC 28, SQCC 10, SCC 4, LCC 1, other 6	*n* = 41	Cell-free proteins from BALF supernatant	Liquid chromatography–mass spectrometry	Different spectral count values from all abundant proteins	LC diagnosis	-
					133 differentially expressed proteins		-
Hmmier 2017 [57]	*n* = 26 (M 13) Mean age: 65 y Type: AC 13, SQCC 13 Stage: I-II 15, III-IV 11	*n* = 16 (M 8) Mean age: 56 y	Cell-free proteins from BALF supernatant	Liquid chromatography–mass spectrometry	267 differentially expressed proteins	AC diagnosis	-
					261 differentially expressed proteins	SQCC diagnosis	-
					292 differentially expressed proteins	Discrimination of SQCC from AC	-
Liu 2021 [58]	*n* = 85 (M 60) Type: AC 32, SQCC 32, SCC 21 Stage: I-II 30, III-IV 42, unknown 13	*n* = 33 (M 20)	Cell-free proteins from BALF supernatant	Lectin microarray	3-lectin (ECA, GSL-I, RCA120) prediction model	LC diagnosis	AUC 0.96 Sens 92% Spec 94%
					4-lectin (DBA, STL, UEA-I, BPL) prediction model	Discrimination of AC from other subtypes	AUC 0.62 Sens 71% Spec 59%
					1-lectin (PNA) prediction model	Discrimination of AC from other subtypes	AUC 0.69 Sens 80% Spec 67%
					6-lectin (STL, BS-I, PTL-II, SBA, PSA, GNA) prediction model	Discrimination of AC from other subtypes	AUC 0.72 Sens 72% Spec 68%
					6-lectin (MAL-II, LTL, GSL-I, RCA120, PTL-II, PWM) prediction model	Discrimination of early from advanced stage	AUC 0.86 Sens 83% Spec 81%
Kwiecien 2017 [60]	*n* = 18 (M 12) Age: 50–81 y Type: AC 4, SQCC 9, NSCLC NOS 4 Stage: I 4, II 11, III 3	-	Immune cells from BALF cell pellets	Antibody-specific flow cytometry	% Tregs, CTLA-4^+^ Tregs (↑)	LC diagnosis (affected vs. healthy lung)	-
Hu 2019 [63]	*n* = 52 (M 29) Age: 39–73 y Type: NSCLC 26, SCC 26	*n* = 20 (M 12) Age: 35–75 y	Immune cells from BALF cell pellets	Antibody-specific flow cytometry	% PD-1^+^ Tph (↓), PD-1^+^ Tfh/Tph (↑)	SCC diagnosis	-
Hu 2020 [64]	*n* = 67 (M 46) Age: 39–75 y Type: AC 18, SQCC 17, SCC 32 Stage: 0–IIIA 39, IIIB-IV 28	*n* = 14 (M 10) Age: 33–71 y	Immune cells from BALF cell pellets	Antibody-specific flow cytometry	Tregs (↑)	LC diagnosis Discrimination of SCC from NSCLC Discrimination of advanced SCC	-
					IL-10^+^ CD206^+^ CD14^+^ M2-like macrophages (↑)	LC diagnosis Discrimination of SCC from NSCLC Discrimination of advanced SCC Overall survival (worse)	-
			Cell-free BALF supernatant	Cytometric bead array	IL-10 (↑)	LC diagnosis Discrimination of SCC from NSCLC Discrimination of advanced SCC Overall survival (worse)	-
Masuhiro 2022 [65]	*n* = 12 (M 9) Age: 55–70 y Type: AC 7	-	Cell-free BALF supernatant	Cytometric bead array	CXCL9 (↑)	Prediction of treatment response (immunotherapy)	-
			Bacterial DNA from BALF supernatant	16S rRNA sequencing	Bacterial alpha diversity (↑), *Proteobacteria* (↓), *Bacteroidetes* (↑)		-
	*n* = 7		Cellular RNA from BALF cell pellets	RNA sequencing	87 genes were upregulated and 28 were downregulated		-
Zhong 2021 [70]	*n* = 12	*n* = 6	Tumor cells from BALF cell pellets	Antibody-specific immunostaining + fluorescence in situ hybridization	Circulating tumor cell count ≥2	LC diagnosis	Sens 75% Spec 100%
Schmid 2015 [72]	*n* = 26 (M 16) Mean age: 60.2 ± 8.3 y Type: AC 20, SQCC 6	*n* = 21 (M 13) Mean age: 64.7 ± 8.4 y	Cell-free BALF supernatant	BLEIA	ATP, ADP (↑)	LC diagnosis	-
			Cellular RNA from BALF cell pellets	mRNA-specific reverse transcription-PCR	CD39 (↑)	LC diagnosis Discrimination of metastatic disease	-
					P2X4, P2X7, P2Y1 (↑)	Discrimination of metastatic disease	-
Callejón-Leblic 2016 [73]	*n* = 24 (M 16) Mean age: 66 ± 11 y	*n* = 31 (M 23) Mean age: 56 ± 13 y	Cell-free metabolites from BALF supernatant	Direct infusion mass spectrometry	Carnitine	LC diagnosis	AUC 0.88
					Adenine		AUC 0.83
					Choline		AUC 0.78
				Gas chromatography–mass spectrometry	Glycerol		AUC 0.89
					Phosphoric acid		AUC 0.79
Callejón-Leblic 2018 [75]	*n* = 24 (M 20) Mean age: 65 ± 13 y Type: NSCLC 22, SCC 2	*n* = 31 (M 27) Mean age: 54 ± 14 y	Cell-free elements from BALF supernatant	Inductive coupled plasma mass spectrometry	Mn	LC diagnosis	AUC 0.75
					V/Cu ratio		AUC 0.76
Suresh 2019 [94]	*n* = 18 (M 13) Type: NSCLC 15, other 3	-	Immune cells from BALF cell pellets	Antibody-specific flow cytometry	% PD-1^hi^/CTLA-4^hi^ Tregs (↓), % IL-1RA-expressing B cells (↓), % central memory T cells (↑), % CD8^+^ TNF-α^hi^ T cells (↑), % IL-1β^hi^ monocytes (↑)	Prediction of CIP development	-
			Cell-free BALF supernatant	V-plex immunoassays	IL-1β (↓), IL-8 (↓), MIP-3α (↓), IL-12p40 (↑), IP-10 (↑)		-
Crohns 2010 [104]	*n* = 36 (M 29) Age: 47–82 y Type: AC 1, SQCC 33, SCC 2 Stage: I 1, III 18, IV 17	*n* = 36 (M 16) Age: 18–75 y	Cell-free BALF supernatant	ELISA	Il-6 (↑)	LC diagnosis	-
					IL-8 (↑)	Overall survival (worse)	-
Yamagishi 2017 [105]	*n* = 22 (M 16) Type: AC 8, SQCC 8, SCC 4, unknown 2	-	Cell-free BALF supernatant	ELISA	MMP-9 (↑)	Prediction of radiation pneumonitis	-
					VEGF (↓)		-
*Metagenomic biomarkers*							
Wang 2019 [76]	*n* = 51 (M 31) Type: AC 18, SQCC 19, SCC 14	*n* = 15 (M 8)	Bacterial DNA from BALF cell pellet	16S rRNA sequencing	Microbial diversity (↓)	LC diagnosis	-
					*Treponema*		AUC 0.86
Cheng 2020 [77]	*n* = 32 (M 23) Mean age: 64.3 ± 8.4 y Type: AC 16, SQCC 9, SCC 7 Stage: I 7, III 6, IV 19	*n* = 22 (M 12) Mean age: 56.5 ± 14.3 y	Bacterial DNA from BALF supernatant	16S rRNA sequencing	10-genera (*f:Pseudomonadaceae*, *Capnocytophaga*, *Stenotrophomonas*, *Microbacterium*, *Gemmiger*, *c:TM7-3*, *Oscillospira*, *Blautia*, *Lautropia*, *Sediminibacterium*) prediction model	LC diagnosis	AUC 0.79
Lee 2016 [78]	*n* = 20 (M 13) Median age: 64 y Type: AC 13, SQCC 5, SCC 2 Stage: II 6, III 8, IV 6	*n* = 8 (M 7) Median age: 58.5 y	Bacterial DNA from BALF supernatant	16S rRNA pyrosequencing	*Veillonella*	LC diagnosis	AUC 0.86
					*Megasphaera*		AUC 0.78
					Combined panel		AUC 0.89
Patnaik 2021 [79]	*n* = 36 (M 16) Type: AC 24, SQCC 11, other 1 Stage: I	-	Bacterial DNA from BALF supernatant	16S rRNA sequencing	19-genera microbiome signature	Prediction of recurrence after surgery	AUC 0.77
Zheng 2021 [80]	*n* = 32 Type: NSCLC Stage: I-II 23, III-IV 9	*n* = 15	Bacterial DNA from BALF supernatant	16S rRNA sequencing	Differentiated abundance of 19 species	LC diagnosis	-
Jang 2021 [81]	*n* = 11 (M 9) Median age: 63 y Type: AC 8, SQCC 3 Stage: III 5, IV 6	-	Bacterial DNA from BALF supernatant	16S rRNA sequencing	*Haemophilus influenzae* (↓), *Neisseria perflava* (↓), *Veillonella dispar* (↑)	Prediction of treatment response (immunotherapy)	-

BALF, bronchoalveolar lavage fluid; M, male; AC, adenocarcinoma; SQCC, squamous cell carcinoma; NSCLC; non-small-cell lung cancer; NOS, not otherwise specified; SCC, small-cell carcinoma; LCC, large cell carcinoma; EV, extracellular vesicle; PCR, polymerase chain reaction; FRET-PHFA, fluorescence resonance energy transfer-based preferential homoduplex formation assay; PNA, peptide nucleic acid; RIA, radioimmunoassay; ELISA, enzyme-linked immunosorbent assay; BLEIA, bioluminescent enzyme immunoassay; LC, lung cancer; TKI, tyrosine kinase inhibitor; CIP, checkpoint inhibitor pneumonitis; AUC, area under the ROC curve; Acc, overall accuracy; Sens, sensitivity; Spec, specificity. Note: Upward arrows correspond to increased and downward arrows to decreased detection of the biomarker respectively to predict each outcome. Sensitivity is the probability that a test result will be positive when the disease is present. Specificity is the probability that a test result will be negative when the disease is not present. Overall accuracy is the overall probability that a patient is correctly classified. Area under the ROC curve is the probability that a classifier will distinguish between two classes (disease vs. normal).

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
