# Peer review of "Bronchoalveolar Lavage Fluid-Isolated Biomarkers for the Diagnostic and Prognostic Assessment of Lung Cancer"

_diagnostics, 2022, doi:10.3390/diagnostics12122949_

Round 1
Reviewer 1 Report
The article is obviously interesting. However some points should be corrected before the next stage.
1. Key words should be corrected according to MESH.
2. The figures have very low quality and should be improved.
3. The methodology of review is missed. Please describe it briefly.
4. It will be more informative if you will be able to provide at least one summary table about the lung cancer markers with indication of their sensitivity and specificity.
Author Response
We thank the reviewer for his encouraging comments. Our answers are below:
1. We have revised the keywords to match existing MeSH terms.
2. The figures are taken from already published articles with permission from the publisher, so there is no way we can make any alterations to enhance quality.
3. We have briefly described the methodology for the identification of included studies in the Introduction section. The revised text is the following:
This review aims to perform a brief and comprehensive overview of studies published in the previous decade that focus on the utility of BALF for detecting different forms of diagnostic, predictive, and prognostic biomarkers in lung cancer. Studies were identified by searching the MEDLINE/PubMed electronic database for relative reports dating from 2011 to September 2022. The search strategy included a combination of MeSH terms (biomarkers; bronchoalveolar lavage; lung neoplasm) and keywords (biomarker; [prognostic or predictive] marker; [bronchoalveolar or bronchial or lung] lavage; [bronchial or lung] washing; [lung or pulmonary] [cancer or neoplasm or tumor]). According to our judgement, some older seminal articles are also mentioned based on their influence on the afterward-growing literature on the subject.
4. We replaced former Table 1 with a table containing the specific biomarkers with the higher accuracy as diagnostic or predictive markers from all included studies.
Reviewer 2 Report
It is known that the detection of molecular markers in samples during routine bronchoscopy, including many fluid cytology procedures such as bronchoalveolar lavage fluid, can serve as a favorable method for improving the diagnosis of lung cancer. The review includes a description of lung cancer biomarkers that can be measured in bronchoalveolar lavage fluid. However, I lacked a summary table that would summarize data on studies of biomarkers of lung cancer in bronchoalveolar lavage fluid, indicating the number of patients, the histological type of lung cancer, the stage of the disease, as well as sensitivity and specificity. Such data can clearly show what is the most promising in terms of application in laboratory diagnostics. Advantages and disadvantages should also be considered in detail.
Author Response
We thank the reviewer for his valuable comments. We have replaced former Table 1 with a new table summarizing patient characteristics, measurement methods, and accuracy data for the most promising biomarkers from all the studies described in the review. We also enriched the Conclusions section by briefly mentioning advantages and disadvantages of biomarker testing in BALF and the potential for future validation and incorporation in clinical practice. The revised text is the following:
Biomarker testing is a central component in nowadays personalized management of lung cancer patients. Many approved drugs for the treatment of stage IV NSCLC are being prescribed according to the detection and quantification of specific molecular markers, such as EGFR, ALK, ROS1, and BRAF mutations and PD-L1 expression. In the case of minimal or no tumor tissue material available for biomarker testing, BALF has the advantage of containing a wide variety of molecules, either cell-derived or in soluble form, that resemble the tumor molecular profile. Biomarkers with the highest diagnostic, predictive, and prognostic accuracy from the studies reported above are presented in Table 1. High concordance to tissue markers and better accuracy than plasma in oncogene driver detection makes BALF the best alternative for predictive assessment in lung neoplasms. Its main disadvantages are the lower yield of tumor cells or tumor nucleic acids compared to tissue samples, which could impede biomarker test performance, and the higher invasiveness compared to plasma sampling. Furthermore, continuous validation of the methylomic, transcriptomic, and proteomic signatures in BALF is needed so that they may assist in the diagnosis of small pulmonary nodules and guide treatment approaches in clinical practice.
Round 2
Reviewer 1 Report
Dear authors, thank you for the revising and updating the manuscript.
However, some issues are still exist.
1. The conclusion has to be based on the study aim. You should transfer all other thoughts to the discussion section. It is not correct to reference the table in the conclusion.
2. In the table it is not clear why do you used "AUC, area under the ROC curve; Conc, concordance; aHR, adjusted hazard ratio; response rate" as a accuracy measures. It is not clear.
Author Response
We thank the reviewer for his comments. Our answers are below:
1. We separated the final conclusion from the paragraph describing the advantages and disadvantages.
2. AUC is a global measure of diagnostic accuracy and it is mentioned since many studies have not used cut-offs in the biomarkers' levels to calculate sensitivity and specificity. Concordance referred to agreement in biomarker detection between tissue and BALF. We decided to use the term "overall accuracy" instead, which is a classic diagnostic accuracy measure and refers to the overall probability that a patient is correctly classified by the index test. The other measures were removed, since they are not used for accuracy purposes. We also included a note under the table with definitions of these measures.
Reviewer 2 Report
The authors took into account the comments of the reviewer at the previous stage of the review. However, I would like to understand how the studies are arranged in Table 1? I think they should be grouped by biomarker type. Then it will be possible to draw conclusions from the data obtained by different authors for individual biomarkers and compare the sensitivity, specificity and representativeness of the sample. I recommend reorganizing the table.
Author Response
We thank the reviewer for his comments. We grouped studies in the table by biomarker type (genetic/epigenetic, post-transcriptional, post-translational, metagenomic biomarkers), as the reviewer proposed.
Round 3
Reviewer 1 Report
The manuscript was significantly improved.